# Does Keeping Cows for More Lactations Affect the Composition and Technological Properties of the Milk?

**DOI:** 10.3390/ani14010157

**Published:** 2024-01-03

**Authors:** Monika Johansson, Mikaela Lindberg, Åse Lundh

**Affiliations:** 1Department of Molecular Sciences, Swedish University of Agricultural Sciences, P.O. Box 7015, SE-750 07 Uppsala, Sweden; ase.lundh@slu.se; 2Department of Animal Nutrition and Management, Swedish University of Agricultural Sciences, P.O. Box 7023, SE-750 07 Uppsala, Sweden; mikaela.lindberg@slu.se

**Keywords:** cow longevity, milk composition and technological properties, number of lactations

## Abstract

**Simple Summary:**

Swedish dairy cows have an average life expectancy of 5 years, i.e., approximately 2.5 lactations during their lifespan. Increasing cow longevity is associated with better animal welfare and lower greenhouse gases per unit milk and cow. However, it is important that there are no negative effects on milk quality if cows are retained in production for longer periods. This study investigated the composition and technological properties of milk from older (≥3 lactations) and young (1–2 lactations) cows. Apart from higher plasmin and lower plasminogen-derived activity in older cows, the results indicated no major differences in milk quality between the parity groups.

**Abstract:**

This study investigated differences in the raw milk composition and technological properties between cows with different numbers of lactations. In total, 12 commercial herds were visited within a period of 12 weeks. On each farm, milk samples from five young cows (lactations 1–2) and five older cows (lactation ≥ 3) were collected. For each farm, milk samples from the young cows and the older cows, respectively, were pooled. The pooled milk samples were analyzed for gross composition and technological properties. Using principal component analysis (PCA) to assess the overall variation in milk quality attributes and the potential clustering of milk from young cows and older cows, respectively, an effect of breed, but no clear effect of lactation number, was observed. In contrast, one-way ANOVA showed higher plasmin activity (*p* = 0.002) in pooled milk from the older cows, whereas plasminogen-derived activity (*p* = 0.001) and total proteolysis (*p* = 0.029) were higher in milk from the young cows. Likewise, orthogonal projections to latent structure discriminant analysis (OPLS-DA) showed higher plasmin activity in milk from older cows, whereas younger cows had higher plasminogen-related activity and higher total proteolysis. To conclude, except for plasmin and plasminogen-related activities, there were no major differences in the composition and technological properties between milk from older cows and young cows.

## 1. Introduction

Animal longevity, i.e., the length of the productive life of a cow, is considered to be strongly related to animal welfare. Injury, poor health, infertility, or bad temperament often lead to involuntary culling of the animal, mastitis being one of the main reasons for early removal from dairy production [1]. As a consequence of mastitis, somatic cell count (SCC) in milk increases [2]. SCC has thus become an important parameter in milk quality payment schemes [3], as regulated by legislation [4]. Milk payments are often based on specified ranges of SCC, where lower levels merit a price premium and higher levels incur payment deductions. This is not only because of cow health concerns, but it is also due to the higher risk of changes in milk composition with elevated SCC, e.g., increased proteolytic degradation of caseins [5]. Conditions other than mastitis can also give rise to elevated SCC, and it is known to increase with parity [6].

Voluntary culling decisions on dairy cows are often based on economic factors, e.g., replacing a cow because of low productivity or age [1]. Culling a dairy cow in a herd will introduce costs if the herd size is to remain unchanged, mostly due to the costs associated with rearing a heifer to replace the culled cow [1]. Keeping dairy cows for longer could therefore improve the economic performance of dairy farms, reduce the environmental footprint of the milk industry, and assist overall in justifying the sustainable use of animals for food production [7]. In fact, many farmers are not aware of the true cost of rearing dairy heifers. According to [8], it takes a dairy farmer an average of 1.5 lactations, or 530 days, to repay the cost of rearing a heifer to calving. During this period, the heifer is non-productive, contributing to higher costs for the farmer and increased greenhouse gas (GHG) emissions allocated to milk production. Once the dairy cow begins to produce milk, the GHG emissions per unit of milk solids will decrease.

Sustainable dairy production needs to be economically defensible while focusing on how to reduce GHG emissions and maintain good animal welfare. Increased cow longe-vity is commonly associated with increased animal welfare [9], but this is not always the case, since older animals are more likely to develop health problems, e.g., lameness and mastitis. Therefore, it is important that an increase in cow longevity is the result of farmers improving herd management to keep animals healthy and comfortable, in turn improving overall animal welfare status [7]. Moreover, if cows are to be retained in production for longer, it is important that there are no negative effects on the composition and technological properties of the milk. 

This study was part of a larger project exploring the benefits of increasing the longevity of Swedish dairy cows, including the effects on methane emissions and farm pro-fitability, through changes in herd management. The aim in the present study was to compare milk from older cows and young cows in the same herd, taking the effect of cow breed into account, to identify differences with respect to the composition and technological properties of the milk with cow age. To our knowledge, the issue of how cow age and overall milk quality are related has not been investigated previously.

## 2. Materials and Methods

### 2.1. Study Design and Collection of Milk Samples

Farms participating in the study (*n* = 11) were recruited in collaboration with Arla Foods Member Service Division and were located in the provinces of Uppland, Södermanland, and Västmanland in Sweden. For practical reasons, only herds milking their cows in a milking parlor, or a tied system, were invited to participate. The farms had cows of two different breeds, Swedish Holstein (SH) and Swedish Red Breed (SRB). Although the aim was to collect a similar number of milk samples from the two breeds, this was not possible. Out of the 11 voluntary farms, 8 farms had only SRB, 2 farms had only SH cows, and one farm had cows of both breeds. This farm was visited twice to collect milk from young and older individuals belonging to the respective breeds. Farms were visited and milk samples from individual cows were collected during the period September–November 2020. The criteria for selection of cows in the participating herds were based on lactation stage and number of lactations. All cows were in mid-lactation, i.e., at least eight weeks after the last calving and not later than 12 weeks before the planned next calving. In 10 of the herds, milk from five young cows in lactation 1 or 2 and milk from five older cows with ≥3 lactations were collected. In the herd with both SH and SRB cows, milk from only four cows in each parity group (young and older) was collected. Cows fulfilling the criteria were randomly selected as they entered the milking parlor. In total, individual milk samples from 28 SH and 88 SRB cows, comprising 58 samples from young cows and 58 samples from older cows, were collected, always in association with evening milking. 

The fresh milk samples were transported at +4 °C to the Department of Molecular Sciences, Swedish University of Agricultural Sciences (SLU), Uppsala, for characterization of composition and technological properties. At the laboratory, 40 mL subsamples of milk from each of the five young cows sampled in a herd were pooled into one sample, and 40 mL subsamples of milk from each of the five older cows sampled in the same herd were pooled into a separate sample. For the farm with both SH and SRB, 40 mL milk samples from young and older individuals of each of the breeds were collected on two separate occasions, one breed at the time. In total, 24 pooled samples were obtained. 

### 2.2. Milk Sample Preparation

Whole fresh milk was used for analysis of pH, milk gross composition, and production of micro-cheeses. Milk (50 mL sample) was defatted by centrifugation (Sorvall Super T21, Sorvall Products L.P., Newton, CT, USA) at 1864× *g* and +4 °C for 10 min. Defatted fresh milk was used for rheology measurements and ethanol stability tests, while defatted milk for the remaining analyses (plasmin/plasminogen activity, milk protein profile, total proteolysis) was stored at −20 °C until use.

### 2.3. Analysis of Milk pH and Gross Composition

Milk pH was measured using a pH meter (Seven Compact S210, Mettler-Toledo, Stockholm, Sweden) after equilibration of the fresh milk samples for 1 h at room temperature. Milk gross composition was analyzed at the Department of Animal Nutrition and Management, SLU (Uppsala). Total fat, protein and lactose, total solids, density, protein in the whey after rennet coagulation (see section below), saturated fatty acids (SFA), unsaturated FA (UFA), mono-unsaturated FA (MUFA), polyunsaturated FA (PUFA), myristic acid (C14:0), palmitic acid (C16:0), stearic acid (C18:0), and oleic acid (C18:1c9) were analyzed by mid-infrared spectroscopy (Fourier Transform Infrared, FOSS, Hilleröd, Denmark). SCC was determined by fluorescence-based cell counting (Fossomatic, FOSS, Hilleröd, Denmark). 

### 2.4. Micro-Cheese Production and Determination of Casein Content and Curd Yield

Production of the micro-cheeses was performed according to [10,11], with some mo-difications. In brief, whole milk samples (10 g in tared tubes, four replicates per milk sample) were pre-warmed in a water bath at 32 °C for 30 min. After this incubation step, calf rennet (75/25 of chymosin/bovine pepsin, 180 international milk clotting units (IMCU) (Scandirenn, Sacco System Nordic AB, Skurup, Sweden) was added to the milk to obtain a final concentration of 0.18 IMCU/mL. The milk samples were vortexed for a few seconds to ensure even distribution of the rennet. Coagulation of the milk was performed in a water bath at 32 °C for 30 min. The resulting gel was cut vertically using a pre-warmed, designated cross-shaped tool. After another 30 min of incubation at 32 °C, the samples were centrifuged at 1650× *g* and 22 °C for 20 min (Sorvall, Super T21), to separate the milk curd from the whey. Curd yield was determined from the weight difference between the initial milk sample and the expelled whey and expressed as g curd per 100 g milk. The difference between total protein from the initial analysis of milk gross composition and protein content in the whey was used as a measure of the total casein content of the milk. Finally, casein number was defined as the ratio between total casein and total protein content.

### 2.5. Rheology Measurements

The rheological properties of the milk were measured using a hybrid rheometer (Discovery HR-3, TA Instruments, Lukens, NC, USA) with a Peltier plate (Hard Anodized Aluminium with Solvent Trap, 40 mm; Discovery HR-3, TA Instruments, Lukens, NC, USA), with a settings temperature of 32 °C, strain 1%, and frequency 1 Hz. Calf rennet (75/25 chymosin/bovine pepsin, 180 IMCU); Scandirenn, Sacco System Nordic AB, Skurup, Sweden) was added to the pre-warmed skimmed milk (15 min at 32 °C in a water bath) to a final concentration of 0.18 IMCU/mL. The time of rennet addition was recorded as the start time and gel formation was monitored for 20 min. Coagulation properties were me-asured as coagulation time (CT), i.e., time in seconds from the point of enzyme addition until a gel strength of 1 Pa was reached, and as gel strength (Pa), measured 20 min after rennet addition (G20). Each milk sample was analyzed in duplicate. 

### 2.6. Ethanol Stability Test

Ethanol stability was defined as the highest ethanol concentration that could be added to the milk sample without causing visual coagulation of the milk. Equal volumes of milk and ethanol, at ethanol concentrations ranging between 40% and 100% in 2% increments, were pipetted to an Eppendorf tube, vortexed for a few seconds, and incubated for 30 min at room temperature before examination [12]. 

### 2.7. Plasmin and Plasminogen-Derived Activity

Plasmin and plasminogen-derived activity in the milk samples were measured as described by [13]. After thawing the milk, each sample was analyzed in duplicate on 96-well microplates (Sarstedt, Nümbrecht, Germany). In brief, plasmin and plasminogen were dissolved from casein micelles by the incubation of defatted milk with ε-amino-*n*-caproic acid, followed by ultracentrifugation for 1 h (Optima MAX-XP, Beckman Coulter, Inc., Bromma, Sweden) at 4 °C and 100,000× *g*. Plasmin activity was measured in the resulting milk serum using 2.5 mg/mL of a chromogenic substrate pyro-Glu-Phe-Lys-*p*-nitroanilide hydroxy chloride (Biophen CS-41(03), Hyphen BioMed, Neuville Sur Oise, France). Plasminogen, i.e., the inactive precursor of the enzyme, was converted into plasmin after activation with urokinase (49.5 Plough units) to measure total activity, i.e., the sum of plasmin and plasminogen-derived activity. Both total and plasmin activity were measured using a multimode microplate reader (POLARstar Omega, BMG Labtech, Ortenberg, Germany) at 37 °C. Absorbance was recorded every 3 min for 120 min, and activity was expressed as change in absorbance at 405 nm per unit time (ΔA405/Δt). Omega Data analysis software (version 5.50 R4) was used for evaluation of the data. Plasminogen-derived activity was finally calculated as the difference between total activity and plasmin activity.

### 2.8. Total Proteolysis

Total proteolysis was measured by a fluorescamine method, based on the reaction of primary amines of trichloroacetic acid (TCA) soluble peptides and free amino acids with fluorescamine, as described by [14] and modified by [15]. In short, the milk samples were mixed with an equal volume of 24% TCA and kept on ice for 30 min before centrifugation at 16,000× *g* for 20 min at 4 °C (Himac CT 15RE; Hitachi Koki Co., Ltd., Tokyo, Japan). Supernatant (20 μL) was mixed with sodium tetraborate with pH 8, fluorescamine was added, and the mixture was loaded into a 96-microwell plate. After shaking the plate for 20 s at a frequency of 300 rpm, fluorescence was determined with a fluorometer (POLARstar Omega, Ortenberg, Germany) using 370 nm excitation and 480 nm emission wavelength. All measurements were performed 23 min after addition of the fluorescamine, and the extent of proteolysis was expressed as leucine equivalents (eq. mM) based on a standard curve with five different concentrations of 0.1 M L-leucine dissolved in 1 mM HCl (1, 0.75, 0.5, 0.3, and 0.05 mM). Each milk sample was analyzed in triplicate. Omega Data analysis software (version 5.50 R4) was used for evaluation of the data.

### 2.9. Casein and Whey Protein Profiles in Milk

Protein separation was performed with a 7100 capillary electrophoresis (CE) system (Agilent Technologies Co., Santa Clara, CA, USA) using an unfused silica standard capillary, as described by [16]. In short, separations were performed after adding 0.017 M D,L-dithiothreitol (DTT) to the sample buffer on the day of sample preparation to disrupt disulfide bridges in the milk proteins. Milk (300 μL) was mixed with sample buffer (700 μL), incubated at room temperature for 1 h, and defatted by centrifugation for 10 min at 10,000× *g* (Himac CT 15R). UV-vis absorbance at a wavelength of 214 nm was used for detection. Calculation of the relative concentration of individual proteins was based on peak area and expressed as percentage of total integrated area in the electropherogram using Agilent 7100 CE, version Rev.C01.08(210) software.

### 2.10. Statistical Analyses

Minitab 18.1 software (Minitab Inc., State College, PA, USA) and Simca 17.0 software (Sartorius Stedim Data Analytics AB, Umeå, Sweden) were used for univariate and multivariate analysis, respectively. The variation in composition and technological properties of milk samples was explored using ANOVA with the Tukey post hoc test in Minitab. In these analyses, cow age and breed were used as fixed factors. The milk quality attributes and technological properties were normally distributed and used as responses. Two different methods of multivariate analysis were used: principal component analysis (PCA) [17] and orthogonal projections to latent structures discriminant analysis (OPLS-DA) [18]. PCA was performed using unit variance-scaled, autotransformed settings to explore the total variation in the composition and technological properties of milk samples as influenced by the parity group and breed. OPLS-DA was performed to explore the relationships between variables and to investigate differences in the composition and technological properties of milk between young and older cows for each breed. The OPLS-DA model was normalized in units for standard deviation and residuals were standardized. The resulting loading plot was inspected to identify variables (i.e., different milk quality attri-butes), with the largest discriminatory power associated with the investigated responses (i.e., cow age and breed). In all statistical analyses, a confidence interval of 95% was used.

## 3. Results

### 3.1. Composition and Technological Properties of Milk in Relation to Cow Age

Differences in the composition and technological properties of milk between young and older dairy cows, irrespective of breed, were evaluated by one-way ANOVA. The results showed higher plasmin activity (*p* = 0.002) in milk from older cows, while plasmino-gen-derived activity and total proteolysis were significantly higher (*p* = 0.001 and *p* = 0.029, respectively) in milk from young cows. In addition, there was a trend for higher relative concentration of total β-casein in milk from young cows (Table 1). 

The effect of cow age on the composition and technological properties of milk was also evaluated within each breed. For the SRB cows, the only differences observed between milk from older cows and young cows were higher plasmin activity (*p* = 0.022) in milk from older cows and higher plasminogen-derived activity (*p* = 0.016) in milk from young cows. There was also a tendency for higher total proteolysis in milk from young SRB cows (Table 2). 

For the SH breed, milk from older cows had significantly higher plasmin activity (*p* = 0.024), while milk from young cows had higher plasminogen-derived activity (*p* = 0.050). Milk from young SH cows also had a significantly higher relative concentration of SFA (*p* = 0.041), a tendency for a higher content of total fat and total solids, and higher total proteolysis (Table 2).

### 3.2. Total Variation in the Composition and Technological Properties of Milk as Influenced by Cow Age and Breed

In the PCA plot of overall variation in milk quality variables, 29% and 18% of the variation was explained by the first and second principal components (PC), respectively (Figure 1). According to the loading plot (Figure 1A), PC1 was positively associated with milk solids, i.e., total fat and protein and G20, and negatively associated with, e.g., coagu-lation time. PC2 explained the variation to a lower extent, mostly discriminating levels of different fatty acids, but was positively associated with curd yield and strongly negatively associated with plasmin activity. In the score plot in Figure 1B, each dot represents a pooled milk sample, with color indicating cow age, i.e., milk from young and older cows. The PCA plot showed no clustering of milk samples belonging to the same parity group, and thus there was no association between cow age and milk composition or technological properties. In Figure 1C, with dots colored according to breed, the PCA plot showed a tendency for milk samples from SH cows to be negatively associated with PC1, forming a cluster in the lower- and upper left-quadrants. According to the loading plot, the underlying reasons for this included, e.g., longer rennet coagulation time and a softer gel, lower total solids, lower relative concentration of casein, higher relative concentration of total whey proteins, and higher lactose content associated with milk from SH cows.

### 3.3. Variables Contributing to Differences in the Composition and Technological Properties of Milk between Young and Older Cows

While PCA showed no association between cow age and the composition and technological properties of the milk, OPLS-DA was performed to identify the variables contributing most to differences between milk from older cows and young cows. The contribution of the different variables and their association with milk from young and older cows, without considering the effect of breed, is illustrated in Figure 2A, where attributes for which the deviation bars do not exceed the zero axis have a stronger influence. The results showed that plasmin activity was positively associated with milk from older cows, whereas total proteolysis and plasminogen-derived activity were negatively associated with milk from young cows. No other variables included in the investigation showed any significant contribution to the difference in milk quality variables between young and older cows. 

Since PCA suggested an effect of breed, the contribution of the different variables to explaining the differences between milk from older cows and young cows was also evaluated within the respective breed using OPLS-DA. Figure 2B and C illustrate the contribution of the analyzed milk quality variables to differences between milk from older cows and young cows of the SH and SRB breeds, respectively. In milk from SH cows, in addition to the higher total proteolysis and plasminogen, higher total fat, SFA, and total solids were associated with milk from young cows, whereas higher PL was related to milk from older cows (Figure 2B). In milk from SRB cows (Figure 2C), higher plasmin activity was associated with older cows, whereas higher plasminogen-derived activity and higher total proteolysis were associated with milk from young cows. No other variables included in the investigation seemed to contribute to the differences between milk from older and young SH and SRB cows.

## 4. Discussion

This study investigated differences with respect to the composition and technological properties of milk from older cows and young cows, while taking the effect of breed into consideration. If cows are to be retained longer in commercial production, thereby contributing to a more sustainable milk production system, it is important that there are no negative effects on the quality of the milk they produce. 

Many previous studies have reported correlations between number of lactations and SCC, with increasing cell count as cows grow older [19,20]. Ref. [21] analyzed the geometric mean SCC at monthly milk recordings in all Swedish primiparous cows in relation to older cows and compared differences in SCC between cows of the SRB and SH breeds. Their results showed that older cows had higher cell counts than primiparous cows, and also that primiparous SRB cows had better udder health than primiparous SH cows [21]. In contrast, in the present study we observed no difference in SCC between milk from older cows and young cows, although milk from older cows had numerically higher SCC values. Overall, the SCC in milk from older cows was generally low, with mean values of 139 × 10^3^ and 187 × 10^3^ per mL milk for SH and SRB, respectively. Interestingly, ref. [22] found no effect of number of lactations per se on SCC when the infection status of udder quarters was taken into account. Instead, they concluded that higher SCC in milk from older cows probably reflects previous infection processes in the mammary gland. Likewise, in a study by [23], cows that were uninfected for at least three years before the investigation produced milk with low SCC, including cows in older age groups. Ref. [24] investigated the differences between herds delivering bulk milk with consistently high or low SCC and did not observe any differences in age structure between the herds, concluding that the higher SCC in one of the herds was not explained by older cows. Considering the low number of herds in the present study, it was not possible to draw far-reaching conclusions. Moreover, even though the individual cows from which milk samples were collected to create the pooled samples were randomly selected as they entered the milking parlor, there is no guarantee that these cows were representative with respect to SCC. Good udder health and low SCC in general in participating herds could also be a reason for the lack of a significant effect of parity on milk SCC in this study.

While there was no difference in SCC between older cows and young cows, ANOVA consistently showed significant differences in proteolytic activities in the milk, with higher plasmin activity in milk from older cows and higher plasminogen-derived activity in milk from young cows, irrespective of breed. Likewise, the OPLS-DA results suggested that of the investigated variables, only plasmin, plasminogen, and total proteolysis explained the differences between milk from older cows and young cows. Plasmin, the major indigenous protease in milk, is mainly secreted as its inactive precursor, plasminogen, and activation of plasminogen into plasmin is mediated by a complex system of activators (PA) and the inhibitors of plasmin and PA [25]. Two types of PA are known to exist in milk, urokinase-type PA (u-PA) and tissue-type PA (t-PA) [26]. While plasmin, plasminogen, and t-PA are commonly believed to be closely associated with the casein micelle and u-PA with neutrophils, u-PA has also been isolated from the casein micelle [27]. Plasmin activity in milk can reflect several processes in the mammary epithelium, such as mammary involution and epithelial cell proliferation [28], possibly by the release of bioactive peptides resulting from degradation of casein [29]. Plasmin activity can lead to defects in dairy products, such as undesirable texture, bitterness, off-flavors, and a decrease in the quality of the final product, while its controlled activity is essential for the development of certain characteristics of long ripened cheeses [27]. Plasmin and plasminogen activity levels in this study were close to those observed in a previous study in which we compared the activities in bulk milk from herds with robot and conventional milking [15]. Plasmin and plasminogen-derived activity were found to be lower in milk from herds using robot milking, likely explained by a higher milking frequency [15]. In the present study, while all cows were milked twice per day in a tied system or milking parlor, differences in milking frequency did not contribute to differences in plasmin activity.

There may be other reasons for older cows having higher plasmin and lower plasminogen-derived activities in their milk compared with the young cows in this study. Ref. [28] observed significant differences in plasmin activity in milk from cows of different lactations, with relatively low plasmin activity during the first three lactations and higher lactation numbers associated with a significant increase in plasmin activity. On calculating the relative importance of different factors to the total variation in plasmin activity, those authors found that SCC explained most of the variation (22%), whereas lactation stage and lactation number each contributed approximately 10% to the total variation [28]. In the present study, milk from older cows had numerically, but not significantly, higher SCC than milk from young cows, and SCC was always <300,000 per mL, the level below which there seem to be no association between SCC and plasmin activity. A study by [13] on the effect of shortening the dry period on plasmin activity in milk found that when cows with three or more parities were subjected to a shorter dry period (four weeks, compared with the conventional eight weeks), they had higher plasmin activity in their post-partum milk. That study concluded that multiparous cows require a longer time for recovery of the mammary epithelium and that the higher plasmin activity reflected an increased mammary epithelial cell proliferation [13]. One explanation for the higher plasmin activity in milk from older cows in the present study could thus be higher cell proliferation in older cows, not only in early lactation, but in general. This is in agreement with the mechanistic lactation model developed by [30], who concluded that the proliferation rate of mammary secretory cells at parturition, the rate of decay associated with reduction in cell proliferation capacity with time, and the specific death rate of mammary secretory cells are all higher for multiparous cows compared with primiparous cows. 

The ANOVA results also indicated a trend for higher total proteolysis in milk from young cows, i.e., besides proteolysis by plasmin also proteolysis by cellular and bacterial proteases [14,31]. Due to practical limitations, in this study the bacterial load of the milk samples was not analyzed, and it is therefore challenging to explain why milk from young cows, with both lower plasmin activity and numerically lower SCC, was associated with higher total proteolysis. The difference was not significant, but the trend for higher total proteolysis in milk from young cows was consistent, irrespective of breed. Whether this is truly a trend, or just a coincidence due to an insufficient number of samples, needs to be further investigated. The low number of observations in the study also poses a risk that other differences in milk composition between the parity groups went undetected.

## 5. Conclusions

There was a significant and consistent difference in plasmin and plasminogen-derived activity in milk between older cows and young cows in this study, despite the limi-ted number of milk samples analyzed. The consequences of keeping cows longer, resulting in higher plasmin activity in the milk, would probably not be of any practical importance, since the difference was relatively low. No other differences in milk between older cows and young cows were observed. These results suggest that, from a milk quality perspective, increased longevity in dairy cows is feasible, as long as their milk SCC is within acceptable levels. 

## Figures and Tables

**Figure 1 animals-14-00157-f001:**
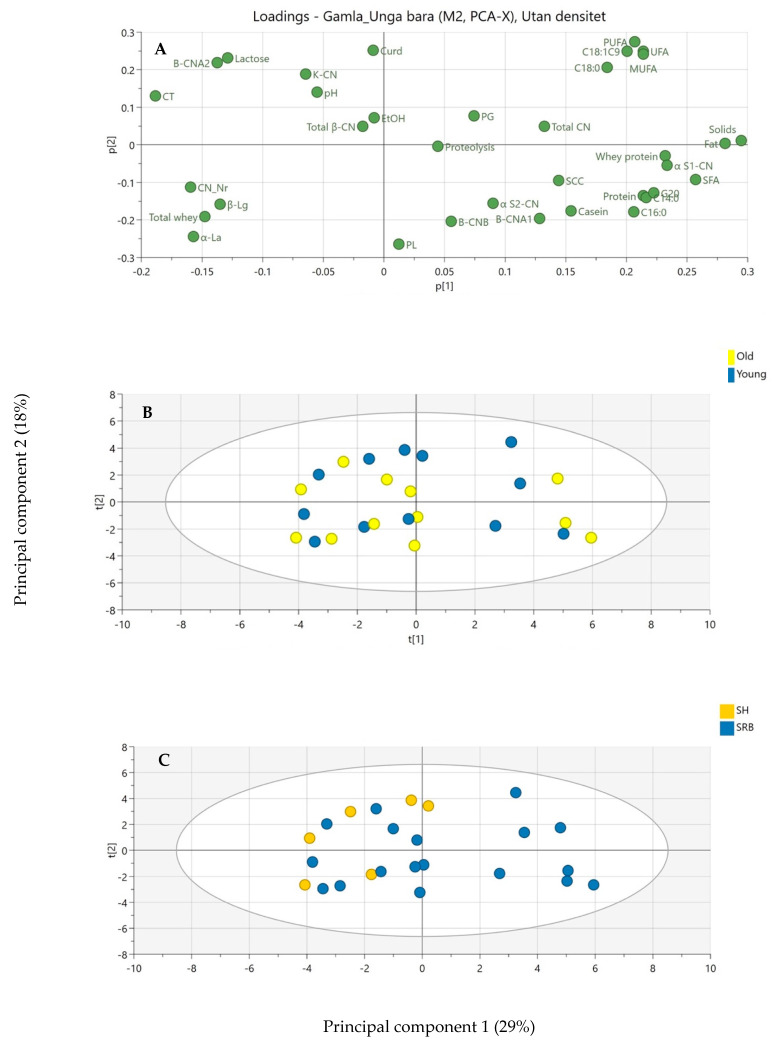
Principal component analysis (PCA) plot showing the overall variation in milk quality variables as influenced by cow age and cow breed. In the score plots, each dot represents a milk sample (*n* = 24) created by separately pooling five individual milk samples from young cows (lactation 1 or 2) and five individual milk samples from older cows (lactation ≥ 3) in 12 herds. The loading plot (**A**) illustrates associations between the investigated traits, with variables grouped together being related. The greater the distance to the origin, the greater the contribution of the variable to the total variation. In the score plots, color indicates (**B**) milk from older and young cows and (**C**) milk from Swedish Holstein (SH) or Swedish Red Breed (SRB) cows. Abbreviations: CN = casein; α-La = α-lactalbumin; ß-Lg = ß-lactoglobulin; Proteolysis = total proteolysis; SFA = saturated fatty acids; UFA = unsaturated fatty acids, MUFA = mono-unsaturated fatty acids; PUFA = polyunsaturated fatty acids; PL = plasmin activity; PG = plasminogen-derived activity; CT = coagulation time; G20 = gel strength at 20 min; CN Nr = casein number; Total whey = whey protein as % of total protein; Whey protein = whey protein fraction after rennet coagulation.

**Figure 2 animals-14-00157-f002:**
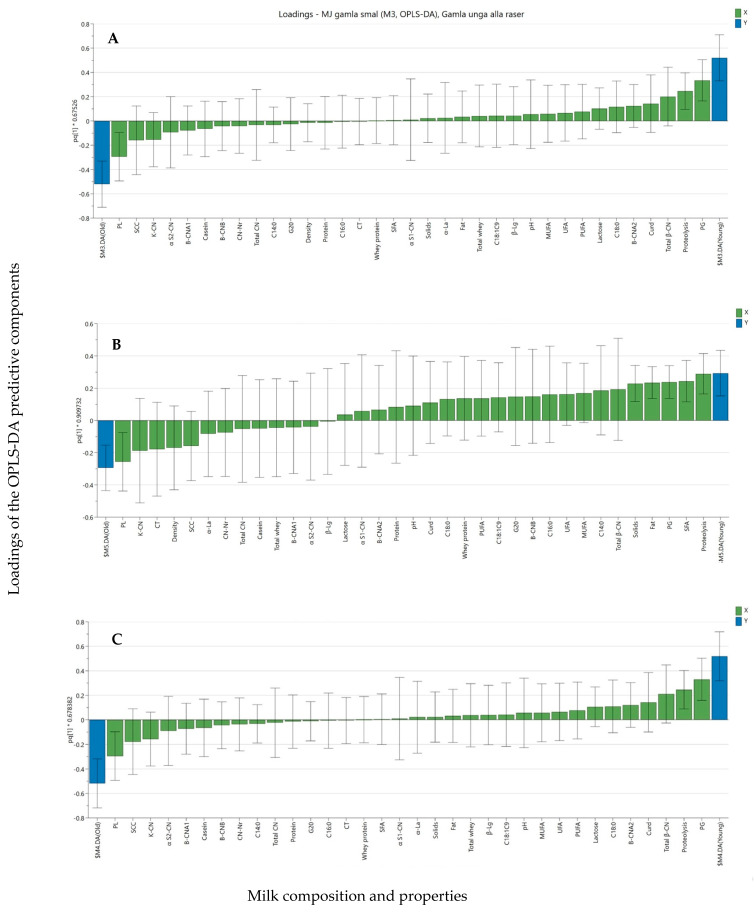
Orthogonal projections to latent structures (OPLS) plot of milk composition and technological properties (green bars) in response to cow age (blue bars: milk from older cows to the left and from young cows to the right), with confidence intervals (95%) for each variable and response. Bar length illustrates the contribution of a variable in explaining the differences between milk from old and young cows, with a taller bar indicating a stronger influence. Milk samples were created by separately pooling five individual milk samples from older cows (lactation ≥ 3) and five individual milk samples from young cows (lactation 1 or 2) in 12 herds, representing either Swedish Holstein (SH) or Swedish Red Breed (SRB) cows. Composition and technological properties (green bars) in response to cow age (blue bars) (**A**) irrespective of cow breed (*n* = 24); (**B**) for Swedish Holstein cows (*n* = 6); and (**C**) for Swedish Red Breed cows (*n* = 18). CN = casein; α-La = α-lactalbumin; ß-Lg = ß-lactoglobulin; Proteolysis = total proteolysis; SFA = saturated fatty acids; UFA = unsaturated fatty acids, MUFA = mono-unsaturated fatty acids; PUFA = polyunsaturated fatty acids; PL = plasmin activity; PG = plasminogen-derived activity; CT = coagulation time; G20 = gel strength at 20 min; CN Nr = casein number; Total whey = whey protein as % of total protein; Whey protein = whey protein fraction after rennet coagulation.

**Table 1 animals-14-00157-t001:** Comparison of the composition and technological properties of pooled milk samples from young cows (lactation 1 or 2) and older cows (≥3 lactations), irrespective of cow breed, collected from commercial farms. Mean value and standard deviation for the different parameters are indicated; *n* = number of pooled milk samples. Statistical significance of differences between milk from young and older cows is indicated by *p*-value, where *p* < 0.05 is considered significant.

	Pooled Milk Samples ^1^ Representing		
Parameter	Young Cows (*n* = 12)	Older Cows (*n* = 12)	*p*-Value
Milk composition (g/100 g)			
Total protein	3.68 ± 0.26	3.69 ± 0.27	0.903
Casein	2.71 ± 0.20	2.76 ± 0.15	0.547
Casein ratio (%)	73.72 ± 1.34	73.98 ± 1.71	0.683
Protein in whey ^2^	0.97 ± 0.08	0.96 ± 0.12	0.953
Total fat (g/100 g)	4.97 ± 0.74	4.88 ± 0.68	0.770
SFA	3.29 ± 0.56	3.28 ± 0.46	0.966
UFA	1.32 ± 0.27	1.26 ± 0.23	0.580
MUFA	1.00 ± 0.23	0.96 ± 0.19	0.626
PUFA	0.13 ± 0.05	0.12 ± 0.04	0.502
C16:0	1.26 ± 0.30	1.50 ± 0.24	0.974
C18:0	0.44 ± 0.14	0.39 ± 0.10	0.286
C19:1c9	0.82 ± 0.20	0.80 ± 0.16	0.748
C14:0	0.63 ± 0.11	0.65 ± 0.10	0.737
Lactose (g/100 g)	4.71 ± 0.13	4.66 ± 0.12	0.385
Total solids (g/100 g)	13.97 ± 0.84	13.92 ± 0.80	0.885
SCC (×10^3^ cells/mL)	107 ± 85	175 ± 112	0.111
pH	6.62 ± 0.05	6.60 ± 0.04	0.520
Protein fractions ^3^ (%)			
Total casein	87.81 ± 2.00	87.89 ± 1.20	0.910
α_s1_-casein	22.68 ± 0.79	22.65 ± 0.98	0.935
α_s2_-casein	7.37 ± 1.24	7.70 ± 0.83	0.451
Total β-casein	50.49 ± 1.55	49.51 ± 0.68	0.057
β-casein B	4.63 ± 0.37	4.71 ± 0.60	0.692
β-casein A1	13.13 ± 7.15	14.80 ± 7.25	0.575
β-casein A2	33.08 ± 7.46	29.99 ± 7.33	0.317
κ-casein	7.38 ± 1.41	8.13 ± 1.28	0.188
Total whey protein ^4^	9.71 ± 1.31	9.55 ± 1.07	0.758
α-lactalbumin	2.14 ± 0.37	2.10 ± 0.33	0.782
β-lactoglobulin	7.56 ± 1.00	7.46 ± 0.78	0.770
Technological properties			
Coagulation time (s)	615 ± 170	610 ± 149	0.938
G20 ^5^ (Pa)	48.81 ± 34.29	54.10 ± 38.30	0.724
Curd yield (%)	61.55 ± 8.53	57.53 ± 7.05	0.221
EtOH stability (%)	81.67 ± 10.26	81.33 ± 9.55	0.935
Proteolytic activities			
Plasmin (U/mL)	3.76 ± 1.40	5.81 ± 1.48	0.002
Plasminogen (U/mL)	88.36 ± 8.37	77.19 ± 6.20	0.001
Total proteolysis (eq. mM leucine)	30.65 ± 4.05	25.16 ± 6.10	0.029

^1^ For 11 of the samples, the pooled milk consisted of milk from five young and five older cows, respectively, from the same breed and farm. For one farm, milk from four young and four older cows, respectively, from the same breed were pooled. ^2^ Protein in whey = protein in whey fraction after rennet coagulation. ^3^ Individual proteins expressed as % of total integrated area in chromatograms. ^4^ Total whey protein = whey protein as % of total protein. ^5^ Gel strength at 20 min. SCC = somatic cell count; SFA = saturated fatty acids; UFA = unsaturated fatty acids; MUFA = monounsaturated fatty acids; PUFA = polyunsaturated fatty acids.

**Table 2 animals-14-00157-t002:** Comparison of the composition and technological properties of pooled milk samples collected from commercial farms from young cows (lactation 1 or 2) and older cows (≥3 lactations) of the Swedish Red and Swedish Holstein breeds. Mean value and standard deviation for the different parameters are indicated; *n* = number of pooled milk samples. Statistical significance of differences between milk from young and older cows is indicated by *p*-value, where *p* < 0.05 is considered significant.

	Swedish Red Breed ^1^			Swedish Holstein ^1^		
Parameter	Young Cows (*n* = 9)	Older Cows (*n* = 9)	*p*-Value	Young Cows (*n* = 3)	Older Cows (*n* = 3)	*p*-Value
Milk composition (g/100 g)						
Total protein	3.73 ± 0.26	3.78 ± 0.24	0.660	3.53 ± 0.23	3.43 ± 0.20	0.584
Casein	2.74 ± 0.19	2.78 ± 0.15	0.629	2.63 ± 0.23	2.69 ± 0.14	0.747
Casein ratio (%)	73.45 ± 1.12	73.54 ± 1.57	0.889	74.53 + 1.86	75.29 ± 1.71	0.627
Protein in whey ^2^	0.99 ± 0.08	1.00 ± 0.12	0.789	0.90 + 0.03	0.85 ± 0.07	0.350
Total fat (g/100 g)	4.95 ± 0.84	5.86 ± 0.67	0.711	5.04 ± 0.45	4.29 ± 0.20	0.057
SFA	3.29 ± 0.65	3.41 ± 0.46	0.667	3.70 ± 0.20	2.88 ± 0.10	0.041
UFA	1.29 ± 0.28	1.31 ± 0.22	0.875	1.40 ± 0.29	1.13 ± 0.23	0.252
MUFA	0.98 ± 0.23	1.00 ± 0.18	0.809	1.08 ± 0.24	0.83 ± 0.18	0.228
PUFA	0.12 ± 0.05	0.12 ± 0.04	0.957	0.14 ± 0.05	0.09 ± 0.05	0.350
C16:0	1.50 ± 0.30	1.55 ± 0.24	0.656	0.40 ± 0.09	134 ± 0.19	0.258
C18:0	0.44 ± 0.14	0.41 ± 0.09	0.554	0.44 ± 0.17	0.33 ± 0.09	0.367
C19:1c9	0.81 ± 0.21	0.84 ± 0.15	0.768	0.85 ± 0.21	0.68 ± 0.17	0.327
C14:0	0.63 ± 0.12	0.67 ± 0.10	0.442	0.64 ± 0.06	0.57 ± 0.03	0.176
SCC (×10^3^ cells/mL)	115 ± 95	187 ± 126	0.190	86 ± 52	139 ± 51	0.269
pH	6.62 ± 0.06	6.61 ± 0.05	0.661	6.62 ± 0.02	6.60 ± 0.04	0.547
Lactose (g/100 g)	4.68 ± 0.14	4.62 ± 0.10	0.350	4.79 ± 0.04	4.78 ± 0.08	0.813
Total solids (g/100 g)	13.97 ± 0.97	14.16 ± 0.76	0.650	13.95 ± 0.37	13.19 ± 0.38	0.067
Protein fractions (%) ^3^						
Total casein	88.12 ± 1.41	87.96 ± 1.20	0.798	86.91 ± 3.51	87.70 ± 1.44	0.737
Total whey protein ^4^	9.80 ± 1.47	9.54 ± 1.24	0.698	9.44 ± 0.79	9.59 ± 0.30	0.773
α_s1_-casein	22.86 ± 0.72	22.92 ± 0.87	0.874	22.14 ± 0.87	22.84 ± 0.95	0.705
α_s2_-casein	7.64 ± 0.59	6.95 ± 0.52	0.261	6.56 ± 2.41	6.97 ± 1.28	0.806
Total β-casein	50.39 ± 1.76	49.46 ± 0.67	0.159	50.79 ± 0.74	49.65 ± 0.82	0.155
β-casein B	4.58 ± 0.42	4.88 ± 0.46	0.174	4.78 ± 0.16	4.23 ± 0.80	0.302
β-casein A1	15.28 ± 6.14	17.09 ± 6.86	0.562	6.69 ± 6.87	7.92 ± 2.39	0.784
β-casein A2	31.00 ± 6.84	27.49 ± 6.68	0.286	39.32 ± 6.41	37.51 ± 2.00	0.664
κ-casein	7.26 ± 1.52	7.64 ± 0.78	0.513	7.75 ± 1.22	9.59 ± 1.49	0.172
α-lactalbumin	2.20 ± 0.40	2.10 ± 0.35	0.584	1.96 ± 0.28	2.10 ± 0.29	0.587
β-lactoglobulin	7.60 ± 1.14	7.45 ± 0.91	0.763	7.47 ± 0.56	7.49 ± 0.12	0.969
Technological properties						
Coagulation time (s)	631 ± 192	577 ± 143	0.504	565 ± 75	709 ± 143	0.201
G20 ^5^ (Pa)	49.20 ± 39.80	67.90 ± 38.6	0.472	47.56 ± 10.91	27.90 ± 26.9	0.306
Curd yield (%)	61.98 ± 9.80	57.58 ± 7.88	0.309	60.24 ± 3.49	57.38 ± 4.97	0.460
EtOH stability (%)	80.00 ± 11.45	79.78 ± 10.5	0.966	86.67 ± 2.3	88.00 ± 3.46	0.795
Proteolytic activities						
Plasmin (U/mL)	3.73 ± 1.58	5.68 ± 1.68	0.022	3.85 ± 0.93	6.22 ± 0.67	0.024
Plasminogen (U/mL)	88.36 ± 9.00	78.68 ± 5.86	0.016	88.34 ± 7.82	72.77 ± 5.84	0.050
Total proteolysis (eq. mM leucine)	30.53 ± 4.58	25.22 ± 6.84	0.089	31.13 ± 0.53	24.92 ± 2.79	0.091

^1^ For 11 of the samples, the pooled milk consisted of milk from five young and five older cows, respectively, from the same breed and farm. For one farm, milk from four young and four older cows, respectively, from the same breed were pooled. ^2^ Protein in whey = protein in whey fraction after rennet coagulation. ^3^ Individual proteins expressed as % of total integrated area in chromatograms. ^4^ Total whey protein = whey protein as % of total protein. ^5^ Gel strength at 20 min. SCC = somatic cell count; SFA = saturated fatty acids; UFA = unsaturated fatty acids; MUFA = monounsaturated fatty acids; PUFA = polyunsaturated fatty acids.

## Data Availability

Data are contained within the article.

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
