# Peer review of "Does Keeping Cows for More Lactations Affect the Composition and Technological Properties of the Milk?"

_animals, 2024, doi:10.3390/ani14010157_

Round 1

Reviewer 1 Report

Comments and Suggestions for Authors

“The aim in the present study was to compare milk from older and young cows in the same herd, taking the effect of cow breed into account, in order to identify differences with respect to composition and technological  properties of the milk with cow age.”

Authors must clarify how breed effect was considered

The title isn’t in accordance with the aim. Consider change “ Processability” with “technological  properties” or explain “ Processability” in introduction.

It will be useful for the reader if Authors clarify if they have information of two breeds on only one herd.

2.1. Study Design

This section is hard to understand.

For instead, considering:

12 herds were sampled (5 young cows and 5 older cows)

In one of the herds, however, milk from only four cows in each age group (young and older) was collected.

Why “24 SH and 84 SRB cows”?

I expected 10 SH and 108 SRB cows…

Maybe a table can help the reader in understanding the Study Design

L96 “2.2. Collection of Milk Samples

Explain the advantage of join five groups samples into one sample

L211 “age of the cows” Authors doesn´t know the age of the cows

L226 “Table 1.” Authors must fundament the advantages of table 1.

Why not consider only “Table 2.”?

L246 “Table 2.

Why didn’t Authors consider an experimental design for a two-way ANOVA with interaction between breed and age group?

Explain the six (3+3) pooled samples for SH (maybe I didn’t understand the Study Design)

Authors must discuss that working with n=3, only huge differences between averages have P<0.05

Author Response

Reviewer no 1

The aim in the present study was to compare milk from older and young cows in the same herd, taking the effect of cow breed into account, in order to identify differences with respect to composition and technological properties of the milk with cow age.

  1. Authors must clarify how breed effect was considered

AU: Milk samples from the two breeds were handled separately, from sampling and pooling of milk from young and older cows, respectively, to the evaluation of data. Knowing that there are significant differences in milk composition between the breeds (causing a larger total variation in milk composition), we wanted to investigate if differences between young and older cows could be distinguished when breed was NOT considered. In this evaluation, the number of observations was larger than when we evaluated within breed. Despite a larger total variation, differences in plasmin/plasminogen and total proteolysis were significant between the age groups (Table 1). Evaluating differences between young and older cows within breed, the number of observations was smaller, but differences in plasmin/ plasminogen and total proteolysis were also this time significant between the age groups (Table 2).

  1. The title isn’t in accordance with the aim. Consider change “ Processability” with “technological properties” or explain “ Processability” in introduction.

AU: The reviewer is right, and it is correct that we have the “technological properties” in mind. Title has been revised. L 2-3.

  1. It will be useful for the reader if Authors clarify if they have information of two breeds on only one herd.

AU: Since we only had 2 farms with only SH cows, we also included one farm with both SH and SRB cows. This farm was thus visited twice, once to collect milk from young and old SH cows, and once to collect milk from young and older SRB cows. In the case of this farm, the pooled milk samples from young and old cows, respectively, consisted of milk from 4 and not 5 individual cows. This is now described in the Material and methods section. L77-84; L91-94; L100-102.

  1. Section 2.1. Study Design. This section is hard to understand. For instead, considering: 12 herds were sampled (5 young cows and 5 older cows). In one of the herds, however, milk from only four cows in each age group (young and older) was collected.
  2. AU. This section has been revised to improve clarity; we agree that it was a bit unclear. We had 11 different farms, but one farm was visited twice (both SH and SRB cows), and therefore 12 herds. L77-84; L91-94; L100-102.

  1. Why “24 SH and 84 SRB cows”? I expected 10 SH and 108 SRB cows…Maybe a table can help the reader in understanding the Study Design

AU: See previous response (4); we agree and apologize that the study design was unclear; it has now been revised. The study was based on 2 SH farms, with 5+5 cows each. In addition, 4+4 samples from SH cows were obtained from the farm with both SRB and SH, in total 28 SH cows. The study also included 8 SRB farms, with 5+5 cows each. In addition, 4+4 samples from SRB cows were obtained from the farm with both SRB and SH, in total 88 SRB cows. Thus, the total number of cows in the study was 116, resulting in the following number of pooled samples, i.e., the milk samples which were analyzed:

SH: 3 pooled samples from cows in lactation 1 and 2, and 3 pooled samples from cows in lactation ≥ 3.

SRB: 9 pooled samples from cows in lactation 1 and 2, and 9 pooled samples from cows in lactation ≥ 3. L77-84; L91-94; L100-102.

  1. Section 2.2. Collection of Milk Samples”. Explain the advantage of join five groups samples into one sample.

AU: From the beginning, we hoped to recruit a much larger number of commercial farms, differing with respect to the average number of lactations of cows in their herds. However, the pandemic made it difficult to recruit voluntary farms. To reduce the effect of the individual variation in milk composition between cows, and “simulate” bulk milk, we decided to pool milk from individuals belonging to the same age/ parity group on each farm. Milk samples from five cows in their 1-2 lactations and from five cows with ≥ 3 lactations were collected on each farm and then pooled to create one sample from young and older cows, respectively. The hypothesis was, if there are differences in bulk milk composition between farms with a higher average number of lactations of cows, and farms with a lower average number of lactations of the cows, these differences would be identified in this study, since our pooled samples either consisted of milk from only young cows or pooled milk from only older cows.

  1. L211 “age of the cows” Authors doesn´t know the age of the cows

AU: This is true, although, cows that are in their 1 and 2nd lactation are older than those in their 3rd and higher lactations. Text has been revised, and “age of the cows and breed” replaced by “parity group and breed of the cows”. L210

  1. L226 “Table 1.” Authors must fundament the advantages of table 1. Why not consider only “Table 2.”?

AU: The reason for including this table is to show that even if we combine the results for SH and SRB cows, and thus increase the variation (still with a low number of observations), the differences in plasmin and plasminogen related activities as well as total proteolysis between cows in their first or second lactation, and cows in their 3rd or higher lactation, remain significant. We believe that this really strengthens our findings. These are the only differences between the parity groups, although there is a tendency for a lower relative concentration of ß-casein in milk from older cows. This, could even be explained by the higher plasmin activity in the same milk.

  1. L246 “Table 2.”
  • Why didn’t Authors consider an experimental design for a two-way ANOVA with interaction between breed and age group?

AU: Conducting an experimental design for a two-way ANOVA with interaction would require a larger sample size. This study was affected by practical constraints because of the pandemic, limiting the ability to recruite the desired number of herds for the experimental design. We analyzed the sample sets separately with respect to breed and parity group. Indeed, the investigation of interactive effects might be interesting in future studies as this is not included in our manuscript.

  • Explain the six (3+3) pooled samples for SH (maybe I didn’t understand the Study Design)

AU: Information about the pooling procedure has been revised, see comment 5 above. L77-84; L91-94; L100-102.

  1. Authors must discuss that working with n=3, only huge differences between averages have P<0.05

AU: This is of course true, and despite this, we get significant differences in the proteolytic activities between the parity groups. There is of course a risk that we, due to the low number of observations, have missed other differences between the groups, and this has been included in the discussion on line L434-435.

Reviewer 2 Report

Comments and Suggestions for Authors

According to the authors, milk from younger and older cows has a similar composition. Cows kept in proper health conditions can be kept in breeding longer which reduces the cost of the breeder. In Poland is practiced longer milking utility of cows.

The research layout carried out clearly, the study carried out responds to the objective set. Results presented in an understandable way. 

In the text, the authors' works were cited 6 times. 

Older literature was used in the article - the most recent article cited is from 2017 and older. More recent literature should be cited.

Author Response

Reviewer no 2

According to the authors, milk from younger and older cows has a similar composition. Cows kept in proper health conditions can be kept in breeding longer which reduces the cost of the breeder. In Poland is practiced longer milking utility of cows.

The research layout carried out clearly, the study carried out responds to the objective set. Results presented in an understandable way.

  1. In the text, the authors' works were cited 6 times.

AU: We don’t understand how the reviewer comes to 6 citations, we can only detect that we cite our work 3 times, i.e., the following publications:

13: de Vries, R.; Brandt, M.; Lundh, Å.; Holtenius, K.; Hettinga, K.; Johansson, M. Short Communication: Influence of Shortening 474 the Dry Period of Swedish Dairy Cows on Plasmin Activity in Milk. J. Dairy Sci. 2016, 99, 9300–9306, doi:10.3168/jds.2016-475 11502.

15: Johansson, M.; Lundh, Å.; de Vries, R.; Sjaunja, K.S. Composition and Enzymatic Activity in Bulk Milk from Dairy Farms with 483 Conventional or Robotic Milking Systems. J. Dairy Res. 2017, 84, 154–158, doi:10.1017/S0022029917000140. 484 Animals 2023, 13, x FOR PEER REVIEW 15 of 15

16: Johansson, M.; Åkerstedt, M.; Li, S.; Zamaratskaia, G.; Lund, Å. Casein Breakdown in Bovine Milk by a Field Strain of 485 Staphylococcus aureus. Journal of Food Protection 2013, 76, 1638–1642, doi:10.4315/0362-028X.JFP-13-112.

All the cited papers were associated to the methods section as follows:

Reference 13. Plasmin and plasminogen-derived activity analysis in the milk samples

Reference 15. Total proteolysis measured by a fluorescamine method

Reference 16. Milk protein profile characterization

  1. Older literature was used in the article - the most recent article cited is from 2017 and older. More recent literature should be cited.

AU: Using the data base Web of Science, we have searched for studies related to changes in composition and technological properties of milk when cows get older (higher parities). Also, we have searched for published work, with potential mechanisms behind the observed higher plasmin activity in milk from cows with higher number of lactations. It seems as if most of the original work was conducted in the past, and not much research in this area was published in more recent years. However, among the references we had also included a recently published review by Dallago, Gabriel M., et al. "Keeping dairy cows for longer: A critical literature review on dairy cow longevity in high milk-producing countries." Animals 11.3 (2021): 808. We have made a new search for literature, using other key words, but have come to the same conclusion.

Reviewer 3 Report

Comments and Suggestions for Authors

Dear Authors,

The evaluated manuscript titled “Increasing Dairy Cow Longevity in Sustainable Milk Production. Does Keeping Dairy Cows for Longer Affect Milk Composition and Processability?” is devoted to milk quality depending on lactation number and breed.

The Authors wrote that no such research has been conducted so far. However, there are many studies available in the literature indicating the quality of milk depending on the age of cows, i.a.,:

Sorensen, Annette, D. Donald Muir, and Christopher H. Knight. "Extended lactation in dairy cows: effects of milking frequency, calving season and nutrition on lactation persistency and milk quality." Journal of dairy research 75.1 (2008): 90-97.

Cielava, L., D. Jonkus, and L. Paura. "Lifetime milk productivity and quality in farms with different housing and feeding systems." Agronomy Research 15.2 (2017): 369-375.

Wilms, J. N., et al. "Fatty acid profile characterization in colostrum, transition milk, and mature milk of primi-and multiparous cows during the first week of lactation." Journal of dairy science 105.3 (2022): 2612-2630.

Sitkowska, Beata. "Effect of the cow age group and lactation stage on the count of somatic cells in cow milk." Journal of central European agriculture 9.1 (2008): 57-62.

Holodova, L. V., et al. "The effect of age on milk productivity and reproductive qualities of dairy cows." IOP Conference Series: Earth and Environmental Science. Vol. 315. No. 2. IOP Publishing, 2019.

Mele, Marcello, et al. "Multivariate factor analysis of detailed milk fatty acid profile: Effects of dairy system, feeding, herd, parity, and stage of lactation." Journal of Dairy Science 99.12 (2016): 9820-9833.

Krol, J., et al. "Lactoferrin, lysozyme and immunoglobulin G content in milk of four breeds of cows managed under intensive production system." Pol J Vet Sci 13.2 (2010): 357-61.

Dallago, Gabriel M., et al. "Keeping dairy cows for longer: A critical literature review on dairy cow longevity in high milk-producing countries." Animals 11.3 (2021): 808.

Longevity was expressed by the number of lactations.

Introduction is appropriate, although it should be noted that the length of use of cows on farms is important for economic, not environmental, reasons.

Title of the manuscript does not correspond to the described results and their description and conclusions. Therefore, Authors should completely change the Title according to the aim of the paper. At this point it is too exaggerated compared to the content of the manuscript. It is not known whether the Authors have “sustainable milk production” because there is no information about the cows’ maintenance and feeding system. Authors selected too small group of cows to conclude about a longevity. Longevity is also related to the cows' maintenance and feeding system, which are not described. The age range (lactation numbers) is too short. What in the case of older animals? It is important in the term of longevity. Unfortunately, Authors omitted the most important parameter - cow efficiency (milk yield). On farms, this is the most important factor, next to SCC, that determines the exclusion of cows from the herd. Indeed, TBC is also missing, as mentioned by the Authors. The following information should be completed in the methodology:

L87 The exception should be explained

L124 What modifications?

L128 How long they were vortexed?

L154 How they were mixed?

L176 .. an equal volume.. – How many?

L180 How long they were mixed?

L192 First author name should be cited form [16].

L193 What sample buffer?

Table titles are incorrect.

Authors analysed only 108 milk samples. There are many other factors influencing the obtained values. Nutrition and breed varied. In this arrangement, 108 animals are not enough. Fatty acids also seem unnecessary in this array. The Conclusion is exaggerated. The manuscript can be transformed into a paper on the influence of age of cows (lactation number) and breed on milk quality, but even in this arrangement the number of animals is too small.

In my opinion, the manuscript should not be published in a current form in Animals.

Author Response

Reviewer no 3

Dear Authors,

The evaluated manuscript titled “Increasing Dairy Cow Longevity in Sustainable Milk Production. Does Keeping Dairy Cows for Longer Affect Milk Composition and Processability?” is devoted to milk quality depending on lactation number and breed.

  1. The Authors wrote that no such research has been conducted so far. However, there are many studies available in the literature indicating the quality of milk depending on the age of cows, i.a.,:
  • Sorensen, Annette, D. Donald Muir, and Christopher H. Knight. "Extended lactation in dairy cows: effects of milking frequency, calving season and nutrition on lactation persistency and milk quality." Journal of dairy research 75.1 (2008): 90-97.
    • AU: Not the same objective: this is a study related to lactation persistency.

  • Cielava, L., D. Jonkus, and L. Paura. "Lifetime milk productivity and quality in farms with different housing and feeding systems." Agronomy Research 15.2 (2017): 369-375.
    • AU: Not the same objective (milk yield in focus) and only SCC was reported.

  • Wilms, J. N., et al. "Fatty acid profile characterization in colostrum, transition milk, and mature milk of primi-and multiparous cows during the first week of lactation." Journal of dairy science 105.3 (2022): 2612-2630.
    • AU: This study focused on detailed FA composition, not overall composition and technological properties of the milk

  • Sitkowska, Beata. "Effect of the cow age group and lactation stage on the count of somatic cells in cow milk." Journal of central European agriculture 9.1 (2008): 57-62.
    • AU: Milk, fat and protein yield reported but not detailed composition and technological properties.

  • Holodova, L. V., et al. "The effect of age on milk productivity and reproductive qualities of dairy cows." IOP Conference Series: Earth and Environmental Science. Vol. 315. No. 2. IOP Publishing, 2019.
    • AU: Milk, fat and protein yield reported but not detailed composition and technological properties.

  • Mele, Marcello, et al. "Multivariate factor analysis of detailed milk fatty acid profile: Effects of dairy system, feeding, herd, parity, and stage of lactation." Journal of Dairy Science 99.12 (2016): 9820-9833.
    • AU: Factor analysis was carried out on 47 individual fatty acids, milk yield, and 5 compositional milk traits (fat, protein, casein, and lactose contents, somatic cell score).

  • Krol, J., et al. "Lactoferrin, lysozyme and immunoglobulin G content in milk of four breeds of cows managed under intensive production system." Pol J Vet Sci 13.2 (2010): 357-61.
    • AU: Effect of breed (Polish Holstein-Friesian, Black-White, and Red-White variety, Jersey and Simental) on the content on antibacterial proteins, i.e., immunoglobulin G, lactoferrin and lysozyme.
  • Dallago, Gabriel M., et al. "Keeping dairy cows for longer: A critical literature review on dairy cow longevity in high milk-producing countries." Animals 11.3 (2021): 808.
    • AU: This paper was already cited in our publication, no. 7 in our reference list.

AU: Using the data base Web of Science, we have searched for studies related to changes in detailed composition and technological properties of milk when cows get older, i.e., at higher parities. Also, we have searched for published work, with potential mechanisms behind the observed higher plasmin activity in milk from cows with higher number of lactations. Of the publications provided by the reviewer, they all have other objectives, and most of them, are related to milk yield or just % fat, % protein or cell count. The paper by Dallago et al (2021) is already in the list of cited papers in our manuscript, i.e., no 7 in our list of references,

  1. Longevity was expressed by the number of lactations.

AU: Yes, this is correct.

  1. Introduction is appropriate, although it should be noted that the length of use of cows on farms is important for economic, not environmental, reasons.

AU: the economy is of course a very important aspect when taking decisions, but we don’t agree that there should not be any environmental aspects, e.g., climate impact. A high recruitment rate, e.g., replacing cows after 1-2 lactations, means that a large proportion of the animals in a herd will be non-producing heifers, that don’t produce milk but still emit methane. This gives a higher carbon footprint per kg of produced milk on the farm compared to a farm where cows are kept for more lactations, and calves are instead used for meat production. Recruitment strategy has large impact on economy and must be based on the conditions of the individual farm, but the strategy applied will not only have impact on economy, but also have an impact on environment/ climate changes.

  1. Title of the manuscript does not correspond to the described results and their description and conclusions. Therefore, Authors should completely change the Title according to the aim of the paper. At this point it is too exaggerated compared to the content of the manuscript. It is not known whether the Authors have “sustainable milk production” because there is no information about the cows’ maintenance and feeding system. Authors selected too small group of cows to conclude about a longevity. Longevity is also related to the cows' maintenance and feeding system, which are not described. The age range (lactation numbers) is too short. What in the case of older animals? It is important in the term of longevity. Unfortunately, Authors omitted the most important parameter - cow efficiency (milk yield). On farms, this is the most important factor, next to SCC, that determines the exclusion of cows from the herd. Indeed, TBC is also missing, as mentioned by the Authors.

AU: The title reflects the fact that increasing the longevity of cows, i.e., keeping the cows longer in production without challenging the milk quality, is associated to a more sustainable milk production, based on the climate impact. To increase the longevity of the cows, management (health, reproduction, feed etc) of the herd becomes critical. These aspects were, however, not in focus of this study, the objective was to investigate if there are differences in the composition and technological properties of milk from cows that are in their first and second lactation and cows that are multiparous.

We have, however, revised and shortened the long title to: Does Keeping Cows for More Lactations Affect the Composition and Technological Properties of the Milk?

  1. The following information should be completed in the methodology:
  • L87 The exception should be explained

AU: on the farm with both SH and SRB cows, milk from 4 individuals, not 5, were pooled. This is now explained on L77-84; L91-94.

  • L124 What modifications?

AU: The modifications are described on L122-126

  • L128 How long they were vortexed?

AU: the samples were vortexed for a few seconds; this has now been added.

  • L154 How they were mixed?

AU: Text has been revised: “were pipetted to an Eppendorf tube, vortexted for a few seconds, and incubated..” L153

  • L176 .. an equal volume.. – How many?

AU: The volume is not critical, just that they are mixed at equal volumes. The detailed procedure is described in reference: 15. Johansson, M.; Lundh, Å.; de Vries, R.; Sjaunja, K.S. Composition and Enzymatic Activity in Bulk Milk from Dairy Farms with 483 Conventional or Robotic Milking Systems. J. Dairy Res. 2017, 84, 154–158, doi:10.1017/S0022029917000140.

  • L180 How long they were mixed?

AU: For detailed information, the reader is referred to the reference.

  • L192 First author name should be cited form [16].

AU: The form to cite references follows the instructions of this journal.

  • L193 What sample buffer?

AU: For such detailed information, the reader is referred to the reference.

  1. Table titles are incorrect.

AU: The table text and the table foot notes have been revised.

  1. Authors analysed only 108 milk samples. There are many other factors influencing the obtained values. Nutrition and breed varied. In this arrangement, 108 animals are not enough. Fatty acids also seem unnecessary in this array. The Conclusion is exaggerated. The manuscript can be transformed into a paper on the influence of age of cows (lactation number) and breed on milk quality, but even in this arrangement the number of animals is too small.

In my opinion, the manuscript should not be published in a current form in Animals.

AU: We are fully aware that the quality attributes of the milk are impacted by so many factors, and that age is just one. However, this is an applied study in collaboration with commercial dairy farms. Our milk samples vary in composition due to differences in farm management, mirroring the variation in raw milk quality that dairy processors are facing.  The limitation in number of milk samples is also mentioned on L372-373 and L434-435.

Reviewer 4 Report

Comments and Suggestions for Authors

General remarks:

I carefully evaluated this paper, and I found only some small problematic parts in the manuscript. However, the topic of this manuscript is very interesting for Animals’ readers.

Detailed opinions:

Introduction:

Please, give more information (into the Introduction section) about plasminogen-plasmin in milk, especially, why are these parameters interesting for the farmers point of milk quality!

Materials and methods

line 137: casein number - why not casein ratio?

Results

Please edit the Tables!

Author Response

Reviewer no 4

General remarks:

I carefully evaluated this paper, and I found only some small problematic parts in the manuscript. However, the topic of this manuscript is very interesting for Animals’ readers.

Detailed opinions:

Introduction

  1. Please, give more information (into the Introduction section) about plasminogen-plasmin in milk, especially, why are these parameters interesting for the farmers point of milk quality!

AU: Since the manuscript was not focusing on plasmin, but investigated a number of quality parameters, we prefer not to focus on plasmin already in the introduction. However, in the discussion, the role of plasmin in the mammary tissue and its effect in milk and cheese, are addressed and we refer to the review by Ismail and Nielsen (2010). The action of plasmin on dairy products was now also added in the discussion part L393-396.

Materials and methods

  1. line 137: casein number - why not casein ratio?

AU: we agree that casein ration is a more common term and we have changed casein number to casein number throughout the manuscript.

Results

  1. Please edit the Tables

AU: We are truly sorry for the format of the Tables; they did not look like this when we submitted the draft. We will interact with the editor of the journal to solve the problem since the journal template seems to make a mess of the tables.

Reviewer 5 Report

Comments and Suggestions for Authors

Dear Authors,

Your article "Increasing Dairy Cow Longevity in Sustainable Milk Production. Does Keeping Dairy Cows for Longer Affecr Milk Composition and Processability" is suitable for publication at Animals under minor revision. See below a list of comments to be applied by you before accepting it.

L1-L3 Replace this title by another one. Try to short it.

L13-L14 Replace 'neg-ative' by 'nega-tive'.

L30-L31 Replace 'technolog-ical' by 'technolo-gical'.

L60-L61 Replace 'longev-ity' by 'longe-vity'.

L77-L95 Add a Table to summarize data from each farm under study (location, number of animals, breed, parity, average BW, average BCS, average TDMI, feeding system, major ingredients of the ration, etc.).

L91-L92 Replace 'col-lect' by 'co-llect'.

L106-L107 Replace 'produc-tion' by 'produ-ction'.

L140-L141 Replace 'Al-uminium' by ?Alu-minium'.

L145-L146 Replace 'Co-agulation' by 'Coa-gulation'.

L206-L207 Replace 'dif-ferent' by 'di-fferent'.

L215-L216 Replace 'attrib-utes' by 'attri-butes'.

L222-L223 Replace 'plasmin-ogen' by 'plasmi-nogen'.

L235 Add SFA, UFA, MUFA and PUFA description to the list.

Table 2 Format table to show one line per row.

L262-L263 Replace 'coag-ulation' by 'coa-gulation'.

L278 Check statement in Figure 1.

L320 Delete one space before 'Confidence'.

L320-L321 Replace 're-sponse' by 'res-ponse'.

L331-L332 Replace 'as-sociated' by 'asso-ciated'.

L341-L342 Replace 'contrib-uting' by 'contri-buting'.

L342-L343 Replace 'neg-ative' by 'nega-tive'.

L351-L352 Replace 'val-ues' by 'va-lues'.

L366-L367 Replace 'signif-icant' by 'signi-ficant'.

L372-L373 Replace 'dif-ferences' by 'di-fferences'.

L375-L376 Replace 'inhib-itors' by 'inhi-bitors'.

L377-L378 Delete one space before 'While'.

L392-L393 Replace 'lac-tation' by 'la-ctation'.

L406-L407 Replace 'prolif-eration' by 'proli-feration'.

L447-L525 Check all references to cite them according to Animals' instructions for authors.

Yours sincerely,

Guest Editor.

Author Response

Reviewer no 5

Dear Authors,

Your article "Increasing Dairy Cow Longevity in Sustainable Milk Production. Does Keeping Dairy Cows for Longer Affect Milk Composition and Processability" is suitable for publication at Animals under minor revision. See below a list of comments to be applied by you before accepting it.

  1. L1-L3 Replace this title by another one. Try to short it.

AU: We have suggested a new title, i.e., Does Keeping Cows for More Lactations Affect the Composition and Technological Properties of the Milk?

  1. L13-L14 Replace 'neg-ative' by 'nega-tive'.
  • L30-L31 Replace 'technolog-ical' by 'technolo-gical'.
  • L60-L61 Replace 'longev-ity' by 'longe-vity'.

AU: The text did not look like this when we submitted the draft; the journal template seems to make a mess of both tables and sometimes the text. We would be happy to take a contact with someone at the Editorial Office to solve these problems.

  1. L77-L95 Add a Table to summarize data from each farm under study (location, number of animals, breed, parity, average BW, average BCS, average TDMI, feeding system, major ingredients of the ration, etc.)

AU: We don’t have access to all this data, and that would be a very different study. The objective of our study was not to investigate the effect of the different factors on milk composition, but to investigate if there seems to be an effect of parity on detailed milk composition and technological properties. Again, since for sustainability aspects, there is an interest in keeping cows for more lactations and reducing the recruitment rate.

  1. L91-L92 Replace 'col-lect' by 'co-llect'.
  • L106-L107 Replace 'produc-tion' by 'produ-ction'.
  • L140-L141 Replace 'Al-uminium' by ?Alu-minium'.
  • L145-L146 Replace 'Co-agulation' by 'Coa-gulation'.
  • L206-L207 Replace 'dif-ferent' by 'di-fferent'.
  • L215-L216 Replace 'attrib-utes' by 'attri-butes'.
  • L222-L223 Replace 'plasmin-ogen' by 'plasmi-nogen'.

AU: See previous comment, these hyphenations have not been created by us, but by the template and were not in the original draft when submitted. We will be in contact with the journal how to avoid the problem.

  1. L235 Add SFA, UFA, MUFA and PUFA description to the list.

AU: The abbreviations are now explained

  1. Table 2 Format table to show one line per row.

AU: The template is giving us a headache; tables look nice and are in order before we submit them via the template. We will be in contact with the Editorial office how to avoid the problem.

  1. L262-L263 Replace 'coag-ulation' by 'coa-gulation'.

AU: Sorry to repeat but even here the template is against us and the layout of our manuscript. It is improved for now and from our site it looks as it should. Hope it will be the same even after re-submission.

  1. L278 Check statement in Figure 1.

  1. L320 Delete one space before 'Confidence'.

AU: This has been revised.

  1. L320-L321 Replace 're-sponse' by 'res-ponse'.
  • L331-L332 Replace 'as-sociated' by 'asso-ciated'.
  • L341-L342 Replace 'contrib-uting' by 'contri-buting'.
  • L342-L343 Replace 'neg-ative' by 'nega-tive'.
  • L351-L352 Replace 'val-ues' by 'va-lues'.
  • L366-L367 Replace 'signif-icant' by 'signi-ficant'.
  • L372-L373 Replace 'dif-ferences' by 'di-fferences'.
  • L375-L376 Replace 'inhib-itors' by 'inhi-bitors'.

AU: See previous comment.

  1. L377-L378 Delete one space before 'While'.

AU: This has been revised.

  1. L392-L393 Replace 'lac-tation' by 'la-ctation'.
  • L406-L407 Replace 'prolif-eration' by 'proli-feration'.

AU: See previous comment.

  1. L447-L525 Check all references to cite them according to Animals' instructions for authors.

AU: The references were re-checked

Round 2

Reviewer 1 Report

Comments and Suggestions for Authors

The Authors responded satisfactorily to the questions and implemented most of the suggested corrections